# Attributing Air Pollutant Exposure to Emission Sources with Proximity Sensing

**Ricardo Piedrahita [1], Evan R. Coffey [1],\***  **, Yolanda Hagar [2], Ernest Kanyomse [3], Katelin Verploeg [1], Christine Wiedinmyer [4,5], Katherine L. Dickinson [6], Abraham Oduro [3] and Michael P. Hannigan [1]**

1   Department of Mechanical Engineering, University of Colorado Boulder, Boulder, CO 80309, USA
2   Department of Applied Mathematics, University of Colorado Boulder, Boulder, CO 80309-0526, USA
3   Navrongo Health Research Center, P.O. Box 114, Navrongo, Upper East Region, Ghana
4   Cooperative Institute for Research in Environmental Sciences, University of Colorado Boulder, Boulder, CO 80309, USA
5   National Center for Atmospheric Science, Boulder, CO 80307-3000, USA
6   Department of Environmental and Occupational Health, Colorado School of Public Health, Aurora, CO 80045, USA
\*   Correspondence: evan.coffey@colorado.edu; Tel.: +1-303-735-5045

**Abstract:** Biomass burning for home energy use contributes to negative health outcomes and environmental degradation. As part of the REACCTING study (Research on Emissions, Air quality, Climate, and Cooking Technologies in Northern Ghana), personal exposure to carbon monoxide (CO) was measured to gauge the effects of introducing two different cookstove types over four intervention groups. A novel Bluetooth Low-Energy (BLE) Beacon system was deployed on a subset of those CO measurement periods to estimate participants' distances to their most-used cooking areas during the sampling periods. In addition to presenting methods and validation for the BLE Beacon system, here we present pollution exposure assessment modeling results using two different approaches, in which time-activity (proximity) data is used to: (1) better understand exposure and behaviors within and away from homes; and (2) predict personal exposure via microenvironment air quality measurements. Model fits were improved in both cases, demonstrating the benefits of the proximity measurements.

**Keywords:** exposure; carbon monoxide; cooking; time-activity; proximity

## 1. Introduction

Air pollution from solid fuels for cooking and heating is responsible for an estimated 2.6 million premature deaths globally [1]. Although the main drivers for personal exposure are generally well understood on a global scale, it can be challenging to determine the impacts and relative importance of pollution sources on personal exposure at the local scale. Understanding where, when, and to what extent individuals are exposed to pollution requires technology that is still in development. Cost-effective, temporally-resolved measurements of personal exposure and related parameters (like location and activity) offer valuable information on sources of exposure. Surveys, which are commonly used to assess some of these factors, are prone to bias. In the household energy field, researchers are often interested in measuring how a change in technology (e.g., a cleaner stove) affects personal exposure to air pollution and associated health outcomes [2–4]. This task is complicated by the fact that individuals are exposed to a vast array of pollutants from a range of sources, many of which are beyond the control of the study. The degree to which this affects a particular study depends on the study design (e.g., randomized vs observational) and the particular context (e.g., the contribution of stove-related sources to personal exposure). Even in "gold-standard"

randomized designs, the presence of other exposure sources can contribute to measurement error in the key outcome of interest (exposures related to the intervention), leading to imprecise estimates and difficulty interpreting results. For example, if a randomized cookstove intervention study finds that an intervention failed to reduce personal exposures, this may be because a) stove-related exposures did not decrease, perhaps because the stove did not reduce emissions and/or because of low usage, or b) stove-related exposures declined, but their contribution to overall exposures was too small to detect an effect. These two scenarios have different policy implications and cannot be disentangled without additional data sources.

Source apportionment has been used previously to provide estimates of pollution source contributions to personal exposure. In the household energy realm, Piedrahita and colleagues found decreased exposure to elemental carbon particulate levels among two cookstove intervention groups in Ghana but were surprised when stove usage monitoring and in-field emission testing suggested that changes in behavior and/or other sources beyond the stoves might explain these differences [5–7]. In China, Huang et al. performed source apportionment of $PM_{2.5}$ exposure samples from women cooking with biomass fuels and concluded that not only were a variety of sources identified but there were no relationships between questionnaire-based measurements and source contributions, pointing to the complex spatial, temporal and behavioral patterns that are not captured by these questionnaires [8]. Source apportionment of PM samples, however, has various drawbacks, including the necessity of time-integrated samples (often 24 h or more), challenging and costly sampling and analysis methods, uncertainty, and variability in the chemical profile of relevant pollution sources, making it difficult to pinpoint the importance of sources to exposures. Advances in location-specific, time-integrated filter sampling show promise assigning exposure to pre-determined spatial microenvironments yet fall short of attributing exposure to specific sources [9].

New tools leverage mobile and wireless technology to enable proximity-detection systems, which offer opportunities to improve measurement of air pollution- human health linkages through time-exposure-apportionment. Here, we describe the development of a low-cost method to provide greater confidence in attributing exposure differences to an intervention as well as more fine-grained insights into sources, behaviors and exposures. Based on Bluetooth Low Energy (BLE) Beacon technology, we developed this proximity sensing system using commercially available BLE Beacons to estimate the user's distance to the cooking area (time-activity data) and therein attribute exposure to a source and improve personal-to-cooking area pollution modeling. Since the development and implementation of our system, additional validation work has been conducted using principles of the system in Guatemala [10] with expressed interest in modeling exposure using an array of low-cost sensors [11].

We developed this method within the context of REACCTING (Research on Emissions, Air quality, Climate, and Cooking Technologies in Northern Ghana), a 200-home randomized cookstove intervention study in the Kassena-Nankana (KN) districts of Northern Ghana (November 2013–February 2016). Participants were randomized into four different intervention arms of 50 households each: one group received two locally made rocket stoves (Gyapa), one received two Philips HD4012 LS stoves, one received one Gyapa stove and one Philips stove, and the fourth was a control group, in which households continued use of traditional 3-stone fires (TSFs) and charcoal stoves (these households were given their choice of stoves after the study). Prior publications from the REACCTING project have reported on the study protocol and provided details about the study region and population [12,13], stove usage, $PM_{2.5}$ concentrations for personal, cooking area microenvironment (predominately outdoors), and regional measurements [6,7,14] as well as cookstove emissions and efficiency [5] and urban and rural differences in exposure and stove use [15].

In this paper, we present modeling results for carbon monoxide (CO) exposure as a function of time-activity category using covariates including BLE Beacon proximity data and the intervention groups. The data used are from a subset of all the CO exposure data collected during the study, and are from deployments where the CO data were collected in tandem with the BLE Beacon system.

We also present modeling results relating personal and cooking area microenvironment CO using time-activity data from the BLE Beacon system. The data used in this model were a further subset of the measurements used in the intervention exp model, that also included microenvironment CO measurements. Lastly, BLE Beacon system performance validation results are presented.

*Proximity Monitoring Background*

A Bluetooth Low Energy (BLE) Beacon proximity monitoring system fills a gap in the exposure assessment toolbox. This system, at a high level, is comprised of BLE signal-emitting devices and receiver hubs (e.g., mobile phone) that register and record the received signal strengths, which vary based on distance between the two. This system serves two purposes in this work. First, it provides time-activity information that is used to analyze personal exposure from sources at home vs. sources away from home. Second, by having a prediction of distance from stoves co-located with pollution monitors, it enables estimation of personal exposure from microenvironment (within residential cooking area) measurements. Personal exposure to air pollution has traditionally been difficult to measure due to high costs and power consumption, and participant burden due to factors like instrument size and operating noise [11]. Such issues can lead to non-compliance of protocols by users. Modeling personal exposure from microenvironment measurements is thus an attractive proposition, and past cookstove studies have done this in different ways using time-activity budgets from surveying [16–22]. Baumgartner et al. [22] found a correlation coefficient of 0.58 (95% CI: 0.34, 0.75) between in-home and personal $PM_{2.5}$ concentrations for adult women over 24-hr measurements in rural China, whereas measurements for children were poorly correlated with a correlation coefficient of 0.08 (95% CI: −0.46, 0.59). Cynthia et al. [20] assessed the quality of this relationship for $PM_{2.5}$ in Michoacán, Mexico and found a weak relationship. They noted that exposures from short duration visits in rooms can be important for exposure (like walking into a smoky kitchen for a moment) but can be difficult to record using traditional methods. Patel et al., illustrated variation of $PM_{2.5}$ concentrations on the scale of meters within small residential indoor environments by installing a high density of instruments in test homes burning biomass [23].

Measuring participant time-activity is challenging and current approaches have major drawbacks. Self-reported time-activity measurement approaches [24] are resource intensive and can result in misclassifications [25]. New developments in wireless technologies such as Wi-Fi allow precise indoor location estimates, but resources are required to train the identification system and a high density of Wi-Fi access points is necessary incurring higher costs. Global Positioning System (GPS) devices can be used to assess location [26,27]; however, these tools tend to have a relatively high power consumption and accuracy may suffer in regions with certain geographic characteristics (e.g., mountains, canyons, and dense foliage) and, perhaps more crucially, in indoor environments. Radio Frequency Identification (RFID) tags can be used as binary room-location indicators, but users must place their small 'passive' type badges close to the RFID receiver, making compliance a concern. Larger 'active' RFID badges that use a battery to increase transmission power have been shown to perform well in indoor location testing [28], and would be a viable technology if the additional logging capabilities conferred by the phones were not needed. Cheng and colleagues piloted an ultrasound localization system (Marvelmind Indoor Navigation System) towards mapping indoor $PM_{2.5}$ concentration distributions at very high spatiotemporal resolution (1 s, 1 cm) [29]. Costs are generally higher for both RFID and ultrasonic systems relative to BLE systems.

BLE technology is well suited for indoor localization (i.e., tracking participant movement and proximity to sources) as it offers a simple measurement principle, system flexibility, and commodity pricing for the hardware. Additionally, there is a benefit to the use of a phone as it can be used for additional monitoring tasks, such as acceleration (compliance monitoring, and acceleration-based activity classification [30], real-time data sharing, and GPS which can help identify important non-home pollution source locations. Bluetooth has been studied extensively for indoor localization in marketing applications (e.g., for automated and targeted advertising) but with substantially different goals of high

accuracy and precision using a large number of Bluetooth transmitters in well-characterized spaces. In this work, we show that such a system contributes substantially to personal exposure monitoring with only zonal time-activity information, which is simpler to obtain. We begin by presenting Beacon system performance validation results along with relevant metrics. We then show that distance categorization improves exposure model performance and adds valuable insights about the origins of CO in the context of the REACCTING study.

## 2. Methods

### 2.1. Sampling System Overview

As part of the REACCTING study, a total of 71 48-h personal CO exposure samples were collected, along with BLE Beacon measurements between July 2014 and November 2015. Primary cook females were targeted for participation in the BLE Beacon measurements to better understand the exposures and activity patterns of those spending the most time in cooking areas (Table 1). For 38 of those samples, we also measured cooking area microenvironment CO concentrations (see configurations as shown in Figure 1). In this work, we present results for a subset of personal exposure CO measurements that have corresponding BLE Beacon measurements. Table 1 presents overall sample statistics, and by activity periods, which are defined based on those BLE Beacon measurements. Analysis of the complete personal CO exposure measurement dataset is reported elsewhere [31].

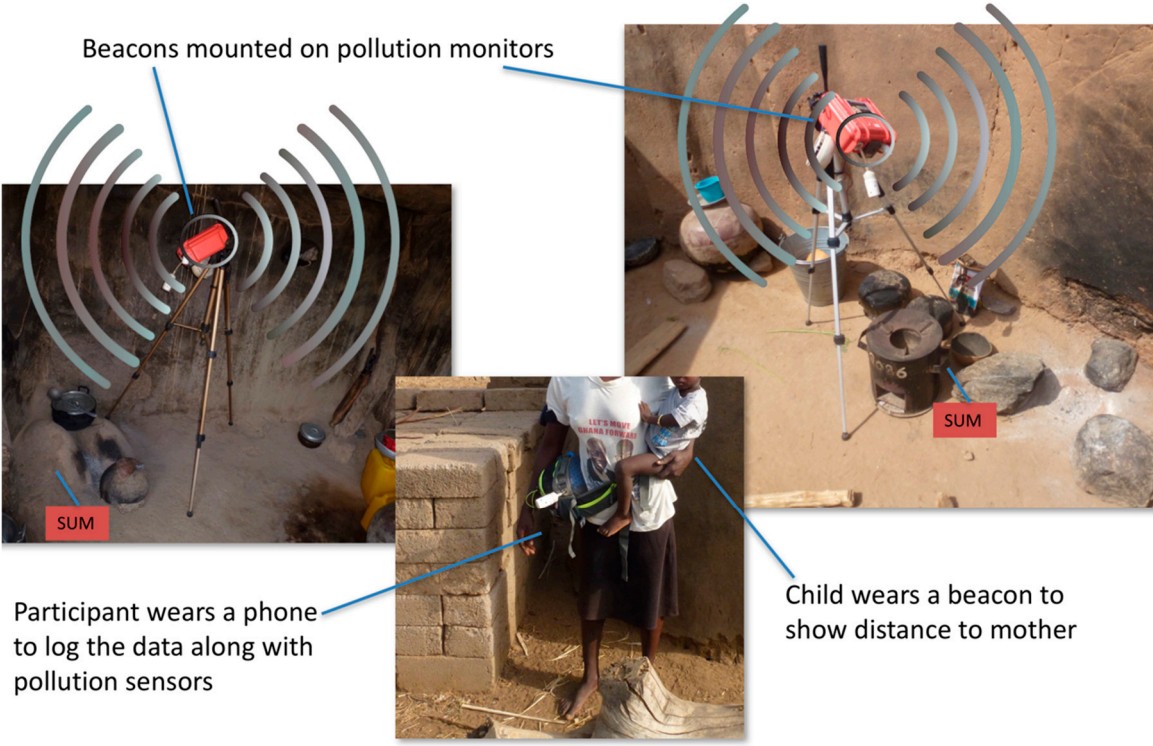

**Figure 1.** Example deployment diagram. Beacons were mounted on the cases of the orange G-Pod pollution monitors. Logging Android phones were worn by participants along with personal air quality monitors.

Microenvironment air quality monitors (G-Pods, Boulder CO, mobilesensingtechnology.com) were placed in the two most-used cooking areas in each home, and were equipped to measure CO and $CO_2$ at sub-minute intervals and averaged to one-minute, and in some cases, integrated $PM_{2.5}$, and total VOCs [7,31]; Supplementary Materials Sections 1 and 2). Many households in this region reported having one or two cooking areas. To capture the most cooking activity, the two most-used cooking areas were monitored. Study participants carried personal CO monitors along with Bluetooth-logging

Android phones. CO calibrations were performed at the University of Colorado-Boulder, as well as in the field, with calibration details presented previously [31]. BLE Beacons were adhered to the G-Pods to provide distance estimates between participants and the two primary cooking areas.

Any personal location tracking device introduces ethical questions, as the potential for misuse and exploitation exists [32]. Therefore, verbal consent was obtained from study participants, after explaining the operation of each instrument they carried. Participants were also given the option to have their data deleted at the end of the sampling period if they so desired.

**Table 1.** Sample statistics for the home monitoring deployments that included BLE Beacons. These include number of deployments, average sampling duration of 48 and 24 h, number of samples removed due to faulty Lascars CO monitors.

| | | All Available Days with Personal CO and Beacon Data | 'Home Cooking by Stove Group' vs. 'Home Not Cooking' vs. 'Away' Data Set (Equation (1)) | Personal vs. Cooking Area CO by Zones (Equation (2)) | Daily Average Personal vs. Cooking Area CO (Equation (3)) |
|---|---|---|---|---|---|
| Duration | Compliant and non-flagged periods deployed | 279 (time-activity periods) | 107 (time-activity periods) | 123 (zone-days) | 38 (days) |
| | Daily compliant duration in hours (mean (SD)) | 19.9 (3.25) | 20.28 (3.6) | 20.94 (3.48) | 20.22 (3.81) |
| | Unique participants | 31 | 22 | 21 | 22 |
| Gender covariates (activity periods) | Primary cook Females | 228 | 101 | 115 | 36 |
| | Non-primary cook females | 51 | 6 | 8 | 2 |
| | Males | 0 | 0 | 0 | 0 |
| | Age of females over 5y (med, SD, max, min) | 38.4, 12.9, 12.3, 73.4 | 39.4, 14.2, 73.4, 12.3 | 39.4, 14.2, 73.4, 12.3 | 39.4, 14.8, 73.4, 12.3 |
| | Age of females under 5y (med, SD, max, min) | 2.1, 0.9, 1.9, 4.2 | 3.3, 0.5, 3.8, 2.9 | 3.3, 0.5, 3.8, 2.9 | 3.3, 0.6, 3.8, 2.9 |
| SES (activity periods) | Poorest | 48 | 24 | 30 | 30 |
| | Poorer | 72 | 24 | 26 | 26 |
| | Poor | 69 | 12 | 15 | 15 |
| | Less poor | 27 | 20 | 23 | 23 |
| | Least poor | 63 | 27 | 29 | 29 |
| Seasons (activity periods) | Harmattan | 150 | 37 | 47 | 13 |
| | Hot dry | 23 | 11 | 15 | 4 |
| | Light Rainy | 31 | 20 | 25 | 4 |
| | Heavy Rainy | 75 | 39 | 36 | 14 |
| | Transition | 0 | 0 | 0 | 0 |
| Stove Group (activity periods) | Control | 31 | 14 | 15 | 6 |
| | Gyapa/Philips | 48 | 27 | 28 | 9 |
| | Philips/Philips | 110 | 29 | 31 | 10 |
| | Gyapa/Gyapa | 90 | 37 | 49 | 13 |

## 2.2. BLE Proximity Sensing System

### 2.2.1. BLE Beacons

BLE Beacons are small battery powered devices that periodically broadcast their media access control (MAC) addresses and other unique identifying information. They have found use in a variety of commercial applications requiring location-based services, such as advertising, where a phone would perform a task upon receipt of a BLE Beacon signal, like offering an in-store coupon. Roximity Model X

Beacons (Roximity, Denver, CO, USA) were used in our study due to their small size (6.4 × 6.4 × 2.5 cm), long battery life of 5 years, and low cost of \$12 (USD) per Beacon. They employ the Apple iBeacon protocol to transmit data. However, only the Beacon MAC addresses and received signal strength indicator (RSSI) are recorded in our application. These signals can be logged with most Bluetooth LE capable systems, including many iOS and Android devices, and a purpose-built Raspberry Pi-based Beacon Logger from Berkeley Air Monitoring Group (Berkeleyair.com, Berkeley, CA, USA). The phones record the identifying information and RSSI from any Beacon within range (generally less than 100 m in open space). RSSI is then converted to a distance measure, providing an open-field distance estimate between phone and Beacon.

### 2.2.2. BLE Beacon Receivers

Phicomm C230w Android phones served as Bluetooth receivers and data loggers (56 USD per unit). A custom Android application was written and installed on each phone to log the Beacon address data, Beacon RSSI, as well as acceleration, GPS, and GPS accuracy (GPS could be manually disabled). Data were logged every six seconds to the phone's micro secure digital (SD) card in the JavaScript Object Notation (JSON) format. The app can be configured to upload data to a remote server, but data were downloaded manually in our study. The phone battery was swapped for an external 6.6 Ah li-ion battery pack, yielding 50–60 h of continuous use. Phones with battery packs weighed 280 g and were consistently placed in the outer pockets of the personal sampling pack.

### 2.2.3. BLE Beacon Data Processing

RSSI is sensitive to path effects like room geometry and obstructions in the measurement area, including people, since water is a strong signal attenuator for Bluetooth that transmits on the 2.4 GHz band. Considering such limitations, many applications use distance categories rather than explicit distance. Here, we used zones defined as 'near' (<15 m), 'medium-near' (15–30 m), 'medium-far' (30–50 m), 'far' (50–90 m), and 'within signal range' (>90 m).

There are two primary modes of localization uncertainty and miscategorization: (1) high frequency attenuation, or a 'teleportation' effect, where phones appear to jump between distances faster than physically probable, and (2) sustained attenuation that consistently places the user farther from the Beacon than they actually are. The first issue can be mitigated with algorithms applied to the data [33,34]. The second issue was not addressed in this work, which may have resulted in bias towards more time spent further from the Beacon, but this effect can be mitigated with more BLE emitters or receivers throughout the study area, or if other types of sensors are also used.

We developed a filtering algorithm to reduce the high-frequency attenuation effects, as previous works have mainly focused on precise within-room location or room categorization [35,36] rather than distance time series categorization. Our approach, the 'maximum velocity' (MV) filter, assigns greater weight to higher signal strength data by defining a maximum change in distance over time (an 'expected walking velocity'), and recursively adjusts the signal strength values $i$ according to the previous value $i-1$. The expected velocity here was set to $\beta = 1$ m/s, and with $\Delta t$ the time between samples, the predefined maximum distance is then $d_{max} = \beta \Delta t$, giving $d_i = d_{i-1} + \beta \Delta t$. For example, as we collected six-second data, if consecutive distance readings are 5m and 25m, the second data point would be modified to $d_2 = 5$ m + 1 m/s × 6 s = 11 m. Median values from each minute are then extracted from this data to further reduce noise and align with other minute-data.

### 2.3. Proximity Calibration and Validation

In July 2016, the Beacon RSSI values were calibrated and validated. We first performed testing in an open field as a base case to assess distance calibration reliability and the effectiveness of our classification scheme. Two phones were placed in the center of a set of concentric circles at radii of 2 m, 5 m, 10 m, 20 m, and 40 m, and a person wearing two Beacons on either hip walked slowly and randomly throughout each zone for 20 min. These zone categories were different (smaller) than

those used in the models for the field data because we found that in the field, sustained attenuation interference effects observed in typical homes resulted in unbalanced distributions over the distance categories, and we wished to balance them. A second validation test was later performed in a location with additional obstructions to mimic characteristics of study households (Supplementary Materials Section 5). In-field calibrations of Beacon RSSI would be beneficial in future work.

Before the start of each 20-min testing period, there were periods when the tester stood still at the intersections of the areas, in order to generate known calibration data. The data from both Beacons and both phones were aggregated to generate a single calibration function for all data collected during validation testing as well as throughout the study deployments in Ghana. Details on BLE Beacon distance calibrations are presented in Supplementary Materials Section 3, Figures 1 and 2.

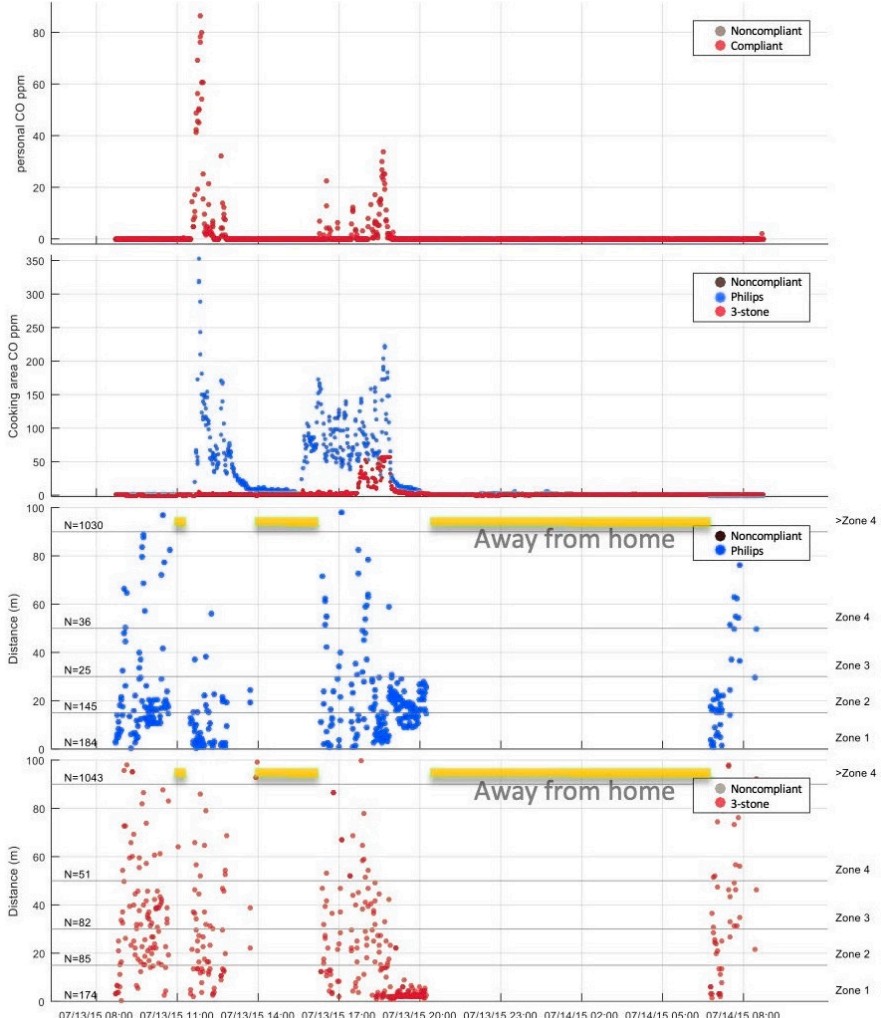

**Figure 2.** Time series showing personal and cooking area CO concentration with Beacon proximity data to each cooking area. The cooking area monitor, and their respective Beacons were named G8 and G9. Lower plots show proximity to the given cooking areas, and the number of samples observed in each zone, or in the case of >zone 4, when signal is weak or lost. Zone thresholds were defined as 15 m, 30 m, 50 m, and 90 m.

As part of the validation we tested the performance of a merged signal that combined the two Beacons worn on the hips, by selecting the stronger signal at each time point, then applying the MV filter. This approach could be used to reduce the multi-path and attenuation effects in future deployments, though if participants carry the Beacons rather than the phone, they would miss out on benefits of carrying the phone, such as GPS and acceleration logging. Classification performance was

assessed using the matching success rate and the rate at which the predicted classification was within one zone of the correct zone, for all available combinations of phones and Beacons.

Validation testing for the 'open field' deployment and all combinations of phones and Beacons, when using the MV filter and data from a single Beacon, showed correct classification of zones on 34.7% of observations, and 65.3% of observations were within one zone of the correct zone. In the later validation test with additional obstructions, those classification rates were 28.2% and 68.6%, respectively. The classification rates when using merged data from both Beacons on the hips were 53.2% and 89.5% for correctly classified and within-one classification respectively for the 'open field' test, and a similar 46.0% and 91.3% for the validation set with obstructions (Supplementary Materials, Figures 3 and 4). Participant compliance, meaning the daytime hours when participants were predicted to be wearing the phone and air sampling equipment, was estimated at 81.9% using the variability in the Bluetooth signals over time ([37], Supplementary Materials Section 5.

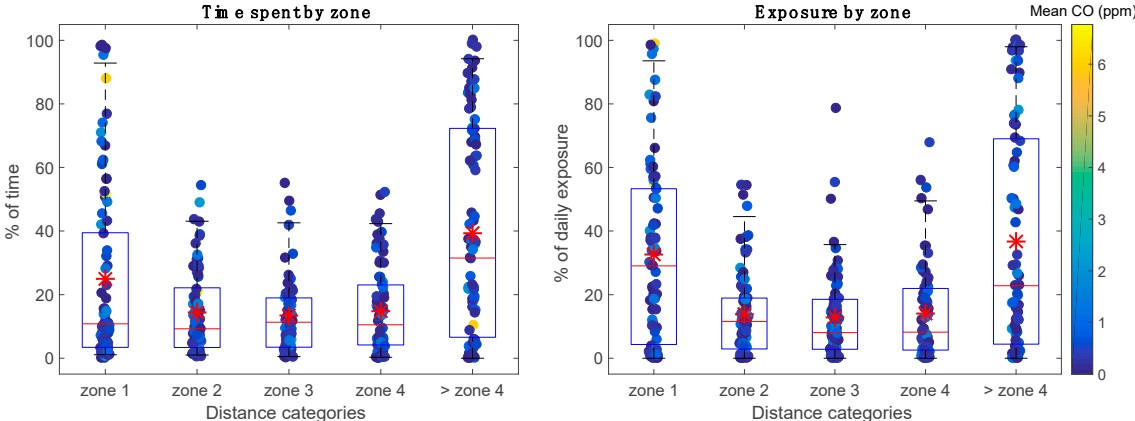

**Figure 3.** Time spent in each zone as a percentage of the day. Marker colors indicate the day's mean exposure to CO. Some participants spent nearly the entire day within zone 1, leading to questions about compliance. Additional sensor streams could improve our measurement of compliance in future work.

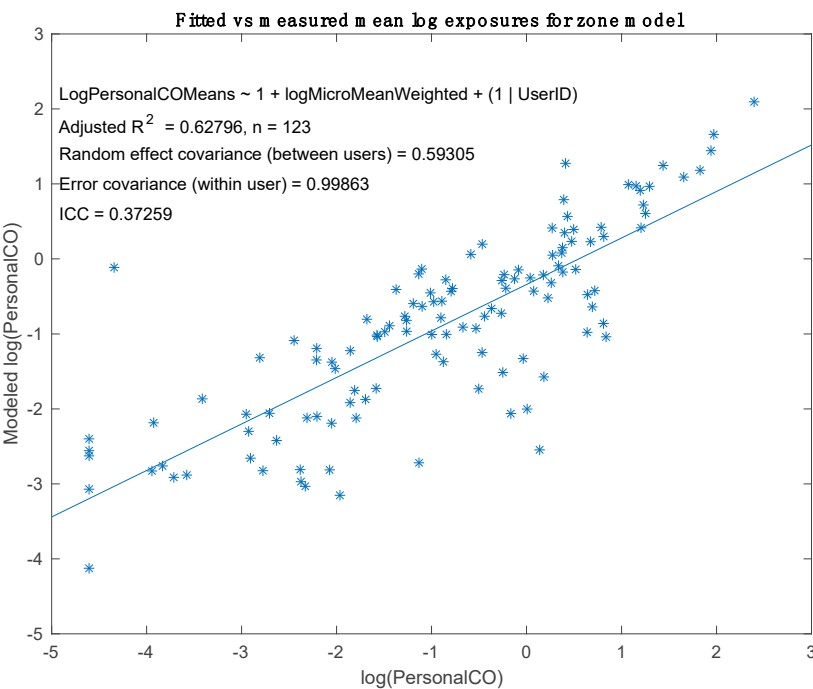

**Figure 4.** Relationship between at-home personal and cooking area CO using Beacon weight-derived cooking area CO, using Equation (3).

*2.4. Attributing CO Exposure as a Function of Time-Activity Category*

The inclusion of time-activity data allows us to gain additional information about how different cookstoves and cooking behaviors impact CO exposure. To help illustrate this, a time series of Beacon proximity data along with personal and cooking area CO concentration is presented in Figure 2. The data show a clear relationship between personal CO exposure and cooking area CO when the user is at home, and sharp reduction of personal CO when the user leaves home at 13:00, as the home CO level remains elevated. This household also appears to use the Philips stove more than the three stone fires (TSFs) throughout the day. This user is unique in that they do not spend the night within range of the cooking area, either spending it in another home, or obstructed enough to be out of range. It can also be seen that periods spent near the stove, as defined by the nearest proximity values, have high variability. It is difficult to discern whether this is due to real movement or teleportation effects. Additional Beacons could reduce this uncertainty in future studies.

$$\text{Log(Mean personal CO}_{ijk}) = \beta_0 + \beta_1(\text{Time-Activity}_{ijk}) + \alpha_j + e_{ijk} \tag{1}$$

A mixed effects regression model was used to determine the effect of the intervention on CO exposure for various time-activity categories. Specifically, in Equation (1), log(Mean personal $CO_{ijk}$) represents the log-transformed average CO concentration on day *i* for individual *j*, in time-activity category *k*. The 'time-activity' categorical variable is defined using the Beacon proximity data in conjunction with available microenvironment CO data, and has six possible states: State 1 ("Away From Home"): participant is considered 'away from home' if more than 90 m away from both cooking areas; State 2 ("Home Not Cooking"): participant is within 90 m of any cooking area but no cooking appears to be in progress; and States 3–6 ("Home Cooking"): for each of the four stove groups, participant is within 90 m of any cooking area in the home, and any cooking area CO measurement in the home is above 10 ppm (Supplementary Materials, Table 1). In other words, the 'home cooking' category is interacted with stove group giving categories of 'control group home cooking' (State 3), 'Gyapa/Gyapa group home cooking' (State 4), 'Philips/Philips group home cooking' (State 5), and 'Gyapa/Philips group home cooking' (State 6). The 10ppm concentration threshold was chosen as it is substantially higher than observed background concentrations in the region, and because temperature-based stove usage data was not available for all households.

Covariates such as socioeconomic status and season were too sparse in some categories to include. The individual random intercept $\alpha_j$ accounts for the correlation within subjects due to repeated measures, and $e_{ijk}$ represents the random variation from subject to subject. More than 12 h of data were required of the daily data (50% data completion) for inclusion in the model, as the primary goal of these models was to assess the system rather than assess exposure over the entire intervention.

This model is useful for identifying time-activity categories with high average exposures. However, it is also beneficial to consider elapsed time spent in each time-activity category to identify where the most exposure comes from each day. Equation (1) was thus modified by changing the dependent variable to total exposure by daily time-activity (ppm-hr) (Equation (2)). This approach highlights nuances in different cooking behaviors. For example, if one stove group has very high average exposures near the stove but does not spend as much time near that stove, total exposure levels may be different than stoves that have the opposite effect.

$$\text{Log(Total personal CO}_{ijk}) = \beta_0 + \beta_1(\text{Time-Activity}_k) + \alpha_j + e_{ijk} \tag{2}$$

Modeling CO Exposure Using Proximity and Microenvironment Monitoring

In addition to the exposure assessment models described in Equations (1) and (2), we also investigated average personal CO exposure as a function of the user's distance away from each cooking area and the cooking area CO measurements (Equation (3)). This approach reflects previous efforts to estimate personal exposures by assigning mean area concentrations from different areas in a home,

using time-location budgets [20,21,37,38]. Such an approach is expected to perform better with precise time-location budgets from a BLE Beacon system. Here, we test whether this is the case when looking at the cooking area microenvironment by isolating exposures when the users are at home. The dependent variable was the log transformed average personal CO exposure for each user deployment *i*, at each distance zone *j* from a cooking area, using only observations when the participant was within zone 4 (based on the BLE Beacon distance data). The independent variable was cooking area CO, linearly scaled by distance zone so as to account for dispersion (e.g., 100% of the cooking area CO was applied if the user was in the nearest zone to the cooking area, and 80% if in the second nearest), and then log transformed. An exponential weighting scheme was also tested to reflect the Gaussian dispersion of CO through the cooking environment, but resulted in no significant difference in performance, likely due to the naturally high variability in the environment. If multiple cooking areas were monitored, we used a weighted average of the cooking areas based on the participant's proximity to each. It is implicitly assumed that concentrations within each zone are uniformly distributed, and average exposures within each zone are independent of one another.

$$\text{Log(Personal CO}_{ijk}) = \ss_0 + \ss_1(\text{weighted cooking area CO}_{ijk}) + \alpha_j + e_{ijk} \tag{3}$$

To understand the impact of the proximity measurements on this modeling approach, we also estimated this model excluding the Beacon proximity data (Equation (4)), and compared model fit between Equations (3) and (4).

$$\text{Log(Personal CO}_{ij}) = \ss_0 + \ss_1(\text{Daily average cooking area CO}_{ij}) + \alpha_j + e_{ij} \tag{4}$$

## 3. Results

### 3.1. Time-Activity Results

For the 38 samples with personal CO, cooking area CO, and proximity data, we find that on average, participants spent 8.5% (±7.9% SD) of their sampling days cooking at home, 51.4% (±30.5%) of their time at home and not cooking, and 40.1% (±32.1%) of their time away (recall that 'away' is defined as beyond zone 4 (90 m), and can still be within signal range). For the data set in which only personal CO and proximity data was collected (33 additional days, presented as 'all available data' in column 1 of Table 1), 24.9% of the day was spent within zone 1, 14.4% was spent in zone 2, 13.4% was spent in zone 3, 14.8% was spent in zone 4, and 39.3% was spent beyond zone 4 (Figure 3). We observed some participants with over 90% of their time spent in zone 1, which seems unreasonable, and could be indicative of non-compliance (i.e., participants not wearing the sampling pack). Additional compliance filtering steps may be appropriate in future work. Time-location variability among users was high, changing with tasks and behaviors depending on household needs.

### 3.2. Exposure by Proximity

Figure 3 shows that 32.6% of total daily exposure (ppm-hr) was experienced within zone 1, 30.7% was experienced in zones 2–4, and 36.7% was incurred beyond zone 4. In other words, roughly a third of exposure was experienced in the immediate cooking area, while another third was experienced in the home but outside of the cooking area, and the final third was experienced outside the home.

Directly analyzing average exposure by zone, the daily median and average exposures were highest in the near-cookstove regions and decreased with increasing distance from the cooking areas, although this decreasing trend was not statistically significant (Figure S5).

### 3.3. CO Personal Exposure Results Using Home vs. Away Categorization

Table 2 presents results estimated from the mixed effects regression model, Equation (1). The reference category (control group at home and cooking) had expected exposure levels of 3.62 ppm (95%

CI: 0.78, 16.75 ppm). During cooking periods at home, relative to the reference group, the Gyapa/Gyapa group had 82.4% lower exposure (0.64 ppm, (0.10, 4.05), $p = 0.07$), the Philips/Philips group had 62.4% lower exposure (1.36 ppm, (0.20, 9.20), $p = 0.31$), and the Philips/Gyapa group had 81.1% lower exposure (0.69 ppm, (0.10, 4.62), $p = 0.23$). While these differences are not statistically significant at conventional levels, their large magnitudes are notable, and effects could be detected with more precision with larger sample sizes. Exposures in the 'home not cooking' and 'away' categories were 95.0% (0.18 ppm (0.03, 0.94), $p = 0.00$) and 96.5% (0.13 ppm (0.02, 0.65), $p = 0.00$) lower than the reference group, respectively.

Modeling integrated exposure by time-activity category (Equation (2)) yielded similar results, indicating that the categories with the highest average exposures were also contributing to most of the personal exposure (Table 2). The Gyapa/Gyapa, Philips/Philips, and Gyapa/Philips homes were respectively responsible for 94.5% ($p = 0.01$), 71.8% ($p = 0.30$), and 92.7% ($p = 0.01$) lower integrated exposures relative to the control group, who experienced 11.6 ppm-hr of integrated exposure (1.69, 79.1), while cooking at home. 'Away' and 'home not cooking' had integrated exposure contributions that were 89.0% and 96.0% lower than the control group's total daily cooking exposure.

### 3.4. Personal CO Exposure Modeling Using Cooking Microenvironment CO

The model from Equation (3) (Supplementary Materials, Table 2), fitting the log of average personal CO exposure with the BLE Beacon proximity sensor zone-weighted cooking microenvironment CO measurements accounted for 63% of within-subject variability. From Equation (3), for every unit decrease of the log-transformed weighted cooking microenvironment CO (as the participant got closer to the cooking area), there was a 173.5% (124.8%, 232.9%) increase in personal at-home expected CO. The random intercept variance was 0.35, and intra-class correlation coefficient was 0.26 (Figure 4).

Model results using Equation (4), which was like Equation (3) but without the Beacon proximity zone-weighted data, indicate that, on a daily average basis, the log of cooking area microenvironment CO is a significant predictor of the log of personal CO exposure ($p < 0.01$), accounting for 28% of within-subject variability ($R^2_{adjusted} = 0.28$). The coefficient on the log of weighted microenvironment CO was 0.81 (CI = 0.40, 1.21), corresponding to a 124.3% (49.9%, 235.6%) increase in personal CO for a one unit increase in the log of the weighted microenvironment CO. Model fit from Equation (4) was poorer compared to the results from Equation (3), that included the zone weighted averages.

**Table 2.** Summary of results from Equations (1) and (2), modeling personal CO exposure by time-activity categories.

| | Average Personal Exposure vs. 'Home Cooking', 'Home Not Cooking', and 'Away' (Equation (1)) | | | | Total Integrated Personal Exposure vs. 'Home Cooking', 'Home Not Cooking', and 'Away' (Equation (2)) | | | |
|---|---|---|---|---|---|---|---|---|
| | Expected value ppm (95% CI) | Coefficient (95% CI) | % change (95% CI) | P-value | Expected value (ppm*hr) | Coefficient (95% CI) | % change (95% CI) | P-value |
| Intercept (control group home cooking) | 3.62 (0.78, 16.75) | 1.29 (−0.24, 2.82) | NA | 0.10 | 11.57 (1.69, 79.07) | 2.45 (0.53, 4.37) | NA | 0.01 |
| Gyapa/Gyapa Home cooking | 0.64 (0.10, 4.05) | −1.74 (−3.58, 0.11) | −82.4 (−97.2, 11.7) | 0.07 | 0.64 (0.06, 6.45) | −2.9 (−5.22, −0.58) | −94.5 (−99.5, −44.2) | 0.01 |
| Philips/Philips Home cooking | 1.36 (0.20, 9.2) | −0.98 (−2.89, 0.93) | −62.4 (−94.4, 153.8) | 0.31 | 3.26 (0.3, 35.89) | −1.26 (−3.66, 1.13) | −71.8 (−97.4, 210.3) | 0.30 |
| Gyapa/Philips Home cooking | 0.69 (0.10, 4.62) | −1.67 (−3.57, 0.24) | −81.1 (−97.2, 27.6) | 0.09 | 0.84 (0.08, 9.28) | −2.62 (−5.01, −0.22) | −92.7 (−99.3, −19.7) | 0.03 |
| Home not cooking | 0.18 (0.04, 0.94) | −2.99 (−4.62, −1.35) | −95.0 (−99.0, −74.2) | <0.01 | 1.27 (0.16, 9.83) | −2.21 (−4.26, −0.16) | −89.0 (−98.6, −15.0) | 0.03 |
| Away from home | 0.13 (0.02, 0.65) | −3.36 (−4.99, −1.72) | −96.5 (−99.3, −82.2) | <0.01 | 0.46 (0.06, 3.59) | −3.22 (−5.27, −1.17) | −96.0 (−99.5, −69.0) | <0.01 |
| | Equation (1) | | | | Equation (2) | | | |
| Random effect by individual variance | 0 | | | | 0 | | | |
| Random error variance | 2.98 (2.28, 3.89) | | | | 4.69 (3.59, 6.14) | | | |
| Adjusted R-squared | 0.20 | | | | 0.08 | | | |
| N | 107 | | | | 107 | | | |

## 4. Discussion

Model results using Equation (1) (Table 2) showed significantly higher average exposures during 'home cooking' in the control group relative to the 'home not-cooking' and 'away' categories. In other words, higher exposures were incurred when home cooking was happening compared to no cooking or when the participant left the cooking area completely. The intervention group differences during cooking compared to the control group during cooking were large and lower, but not statistically significant, potentially due to low sample sizes and high variability, providing suggestive evidence that CO exposures due to cooking may have been lower for participants in the three intervention groups relative to the control group. This is similar to results from the analysis of the complete daily-averaged CO exposure data previously carried out [31]. This analysis showed lower exposure for the three intervention groups, with the largest reduction for the Philips/Philips group at 14.9% ($p = 0.40$), while the Gyapa/Philips group was 0.1% lower ($p = 1.00$), and the Gyapa/Gyapa group was 5.6% lower ($p = 0.78$). Modeling the effect of stove group using daily averages with only the data available when proximity monitoring took place yielded much larger reductions relative to the control group, but the magnitude of effects seen in the results from Equation (1) are likely also a consequence of the small sample size.

Our suggestive evidence of reductions in exposures are particularly notable given that substantial stove stacking (continued use of traditional stoves alongside improved stoves) was observed across all of the intervention groups [6]. The intervention cookstoves also deteriorated (e.g., Philips batteries or fans failed and/or both stove bodies physically degraded) over the course of the intervention, which may account for the modest exposure reduction in the Philips/Philips group here, as the proximity measurements were collected in the latter half of the study period.

The importance of home-level air pollution sources thus appears to be quite substantial. However, average exposures were based on different time durations, and cooking takes up less time than the 'home not cooking' and 'away' categories. Model results using Equation (2) reflect this, and summary distributions presented in Figure 3 indicate that more than a third of average daily exposure was experienced more than 90 m from the cooking area. Source apportionment of $PM_{2.5}$ in the same study showed that two cooking-related sources accounted for a median 15.3% of elemental carbon (EC) and 9.2% of organic carbon (OC) in personal and cooking area concentrations, with other important sources including a biomass combustion source that appeared to be more regional in nature, and vehicular combustion [7]. It should be noted that neither total daily $PM_{2.5}$, nor EC and OC have been found to be well correlated with CO in rural settings in personal or microenvironmental measurements in the region [7,39].

Results from the models using Equations (3) and (4) show that microenvironment CO measurements coupled with cooking area proximity data can substantially improve prediction of personal exposure using area measurements, even in areas with high variability in cooking location, ventilation, and cooking area geometry.

While the system we have developed shows promising results, further testing should be performed to assess performance and limitations in other regions, household member types (especially children), seasons, and with other pollutants. Additionally, the correlation between microenvironmental pollution concentration and personal exposure is likely to vary based on regions and cooking behaviors, so pilot studies should always be performed to determine model coefficients and quality of fit.

Allen-Piccolo et al. [40] introduced an ultrasound-based time-location monitoring platform (UCB-TAMS) for cookstove applications that displayed promising results. Ultrasound has lower attenuation than Bluetooth, thus improving signal consistency in difficult geometries or crowded spaces, but such systems have not come into widespread use. As used in their study, one receiver is placed in each room of interest, and the users wear the ultrasonic transmitters on the outside of their clothing. A receiver and three transmitters were reported to cost $80 when purchased at scale, very similar to the cost of the system presented here. We were unable to perform a direct comparison between the systems due to UCB-TAMS unavailability.

## 5. Conclusions

With a budget of $120 per set of equipment, we were able to add temporally resolved proximity to stove data to improve our understanding of personal CO exposure from REACCTING. There were significant inter-user differences by exposure location, and thus exposure sources. The results presented here demonstrate the ability to more accurately measure CO exposure differences due to the intervention. Such a system could even be used to customize exposure reduction strategies to different types of users. Additionally, this proximity sensing and exposure monitoring system has application in other settings where there is concern for air pollutant exposure, but limited quantitative knowledge about the relative importance of the pollutant sources.

We find that even in the dynamic and predominantly outdoor homes in Northern Ghana, using the proximity data provided reasonably good performance in predicting personal CO exposure. We would expect improved performance in applications with tighter building envelopes, more time spent in the main cooking area, and fewer sources of combustion emissions. This exposure assessment technique has broad applicability across exposure monitoring domains-ranging from evaluating global health with respect to development to occupational and industrial safety. In addition to improved exposure modeling and the ability to attribute exposure based on proximity to sources, variability of exposure within individuals can be explored in detail; through this approach, behaviors that result in relatively higher exposure, for a given individual, can be determined. While so much information may be difficult to synthesize, in large part because of the high variability, if a study can collect large sample numbers then having such data provides great potential to fully understand the effectiveness of interventions as well as develop behavior- focused exposure mitigation strategies.

**Supplementary Materials:** The following are available online http://www.mdpi.com/2073-4433/10/7/395/s1. Figure S1: RSSI-to-distance calibrations for various calibration models. The bold black line shows a fit using aggregate data from both phones, and both beacons, while the thin lines are phone/beacon specific. Box and whisker plots show the distributions of the all the raw data, with whiskers representing 5th and 95th percentiles. Note that the outlying curves on the top and bottom of the plot are from phone 4, suggesting a performance issue with that phone. Figure S2: Modeled categories vs. known categories for all merged beacon signal data. Percentages add up to 100 by column, as the x-axis represent the known category values. Figure S3: Performance from the validation deployment in an open field. Light colored boxes show the match rate, and dark boxes show the rate at which the algorithm predicted within one zone of the correct zone. Left frames show performance by distance zone, while right frames show overall performance. Top frames show match rates using the MV algorithm, the middle frames show rates using minute medians, and the bottom frames show match rates using the merged beacon data along with the MV algorithm. Figure S4: Performance from the test deployment with additional obstructions. Light colored boxes show the match rate, and dark boxes show the rate at which the algorithm predicted within one zone of the correct zone. Left frames show performance by distance zone, while right frames show overall performance. Top frames show match rates using the MV algorithm, the middle frames show rates using minute medians, and the bottom frames show match rates using the merged beacon data along with the MV algorithm. Figure S5: Mean exposure distributions categorized by zones. Marker colors indicate the participant's average exposure from the entire day, and red stars represent means by zone. Slope of decreasing average exposure by zone was not found to be statistically significant by univariate linear regression. Table S1: Summary of time-activity states and defining criteria used in Equations 1 and 2. Table S2: Summary of results from modeling personal CO exposure by cooking area CO.

**Author Contributions:** Conceptualization, R.P., E.R.C., C.W., K.L.D. and M.P.H.; Formal analysis, R.P., E.R.C. and Y.H.; Methodology, R.P., E.R.C. and Y.H.; Project administration, E.R.C., E.K., C.W., A.O. and M.P.H.; Software, R.P. and E.R.C.; Validation, R.P. and K.V.; Visualization, R.P.; Writing–review & editing, R.P., E.R.C., Y.H., C.W., K.L.D. and M.P.H.

**Ethical Considerations:** This work was reviewed and approved by the Institutional Review Boards at the University of Colorado Boulder, the National Center for Atmospheric Research, and the Navrongo Health Research Center, a part of the Ghana Health Service.

**Funding:** This work was funded by a grant from the US National Science Foundation (Award #83542401) and developed under Assistance Agreement No. 1211668 awarded by the U.S. Environmental Protection Agency to Michael P. Hannigan. It has not been formally reviewed by EPA. The views expressed in this document are solely those of the authors and do not necessarily reflect those of the Agency. EPA does not endorse any products or commercial services mentioned in this publication. The National Center for Atmospheric Research is operated by the University Corporation for Atmospheric Research under the sponsorship of the National Science Foundation.

**Acknowledgments:** Thank you to all the staff and field workers at the Navrongo Health Research Center, for their hard work and dedication, without which this study would not have been possible. Thank you also to Kevin Owens and Cole Richards from Roximity for their guidance and support using the Beacons, and Kun Li from Athlete Architect for his Android App development work. Publication of this article was funded by the University of Colorado Boulder Libraries Open Access Fund. A special thanks goes to Natalie Banacos for her editorial administration.

**Conflicts of Interest:** The authors declare no conflict of interest.

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
