# Peer review of "Attributing Air Pollutant Exposure to Emission Sources with Proximity Sensing"

_atmosphere, doi:10.3390/atmos10070395_

Round 1

Reviewer 1 Report

This study adopted an innovative Bluetooth Low-Energy (BLE) Beacon system to estimate participants’ distances to their cook stoves. The authors conducted calibration models to validate their measurements and also conducted exposure assessment models that incorporated the novel proximity data to improve overall fit. This is a very impressive evaluation and a great step in the right direction for household air pollution studies.

I have a few comments and suggestions to improve the overall readability of the current write up. While the manuscript reads while in most places, at times it is a bit unclear to the reader what is being referred to.

Page 1, line 39: “it can be challenging to determine the impacts and relative importance of pollution sources on personal exposure at the local scale” How can it be challenging, explain further?

Page 2 line 44-onwards: Please rewrite this paragraph. It is unclear which results will be presented as it currently reads.

Lines 44- 46 read as rather oddly since BLE data and CO data from the subset of 38 subjects are presented, but the entire data set of 71 are not. But the subsetted data are also a reflection of the objectives of the n=71 study

Page 3 line 30: certain geographic characteristics such as...?

Page 3 Line 25 to 39 is pretty informative. However, the information could be presented in a reader friendly manner. For instance when BLE were introduced in this section, the reader was made aware from the intro sentence. But when self-reported time activity measurements are introduced, it was unclear where the authors were going. Then GPS was mentioned.

I suggest easing the reader into understanding what this section is about, namely we are reading on the background/literature of proximity monitoring. So when each type/example is under discussion, try to introduce it similar to how BLE was introduced in the first section. Or better still, provide these examples as weaker forms of measuring proximity, then build the case for how the authors developed the BLE for superior proximity measurements in the final paragraph

Page 4 line 11: Is there a reason for 38? Was it a convenience sample?

Page 4 Table 1 Female over 5y: does this mean 38.4 females were over 5 years or the females over 5 years were 38.4 years? I believe tables should be stand-alone from the text. If I were to just read the tables, I would be a bit perplexed about how to determine the units of the numbers on this row and the one below (females under 5 years)

Page 5 Line 3: Why two of the most used cooking areas? (why not three or just the most used cooking area)?

Page 6 Line 35: In which direction did it bias results and how does this affect the interpretation of the current results presented?

Page 7 Line 13-17: I commend authors for this foresight. One question this brings up were the calibrations performed in Ghana? (the ones in Lines 4-12) if yes, were the geographic characteristics of the field and home taken into account? If no, please explain further since I suspect that the terrain in Colorado is different from the terrain in Ghana.

Page 9 lines 7-8: “and any cooking area CO measurement in the home is above 10ppm”…is this 10ppm a 30 second measurement? Or 1 min? 1hr or mean 24hr/48hr CO?

lines 7-8 on page 10 seem redundant since the authors have presented this information a few times already (eg page 4, etc) and this subset of data is the focus of the manuscript

Page 10 lines 14: I am assuming this is + or - standard deviation? (Lines 14, 15)

Page 11 line 12: I am assuming this is 95% CI?

Page 13 line 1-2: I am assuming this sentence pertains to model 4.

General comment: Did the study team conduct baseline measurements prior to deploying the intervention? If yes, was the BLE system employed during that phase? This would have generated some pretty novel data. Since one may be able to tease out changes in behavior due to the new stove. As it stands, can the authors comment on whether the female cooks changes their behavior in any way after the received new stoves? (eg more time in the kitchen or cooking premises, just to acquaint themselves more with the new device)?

Reviewer 2 Report

Dear authors,

I commend you on a well written paper and the originality of the idea to integrate bluetooth as a means of activity based monitoring combined with exposure assessment. It is really an interesting idea.

I have some minor recommendations for changes, which are as follows:

Page 3 lines 11-13 - please include some citations

Page 3 line 40 - comma after "i.e."

Page 4 Table 1 - can probably just remove male gender covariate and state no males

Page 4 Table 1 - I am slightly confused by how to interpret the column Home cooking by stove group' vs. 'home not cooking' vs. 'away' data set (Eq. 1). For example, there were 20.28 daily compliant hours on average for.....not really sure how to interpret this given the column name. Recommend rephrasing

Page 6 line 33-37 - You state that the second issue, "sustain attenuation...", could result in more bias but was not addressed. This is an area of concern for me, but can be mitigated if you were to add a few citations clarifying that while it may be an area of concern, it is not one that is going to invalidate your results.

Page 8 Figure 2 - Color schemes for noncompliant vs. G9 are very similar and extremely hard to distinguish. Please be kind to those of us that are partially red/green colorblind and choose more distinctive color schemes.

Page 9 Lines 3-11 - The different states are very hard to follow in text. Suggest putting these definitions in a table as well to make it more reader friendly.

Page 11 section 3.3 - Please state initially that this is expected mean (95%CI) as it wasn't clear whether this was 95% CI or something else without referring to the table.

General comment - was any assessment done according to fuel type? If so, or if not, do you think this would impact exposure? If not assessed, please make this as a limitation/recommendation, particularly since you also briefly mention source apportionment.

General comment - Any thoughts regarding infiltration from other homes as a contribution to exposure? Several studies have noted this as an issue, perhaps something else to consider in recommendations for future studies?

General comment - It is well established in the cookstove community that exposure being close to a fire is going to be much higher than being away, which I know you are well aware of. The interest level in the cookstove community will be very limited in terms of these findings. That said, I suggest you take the opportunity in your summary and conclusions to emphasize how the expsoure assessment technique (i.e., utilization of bluetooth as a means to track activity in the developing context) is something that can be used in very many different contexts and has much broader applications, which you have done a great job illustrating that through this study. Just a suggestion on trying to think of ways of how you could showcase this work to have broader applications.

Upon addressing these changes, I recommend publication.

Best of luck.
